# Bio-Guided Isolation of SARS-CoV-2 Main Protease Inhibitors from Medicinal Plants: In Vitro Assay and Molecular Dynamics

**DOI:** 10.3390/plants11151914

**Published:** 2022-07-24

**Authors:** Hossam M. Abdallah, Ali M. El-Halawany, Khaled M. Darwish, Mardi M. Algandaby, Gamal A. Mohamed, Sabrin R. M. Ibrahim, Abdulrahman E. Koshak, Sameh S. Elhady, Sana A. Fadil, Ali A. Alqarni, Ashraf B. Abdel-Naim, Mahmoud A. Elfaky

**Affiliations:** 1Department of Natural Products and Alternative Medicine, Faculty of Pharmacy, King Abdulaziz University, Jeddah 21589, Saudi Arabia; gahussein@kau.edu.sa (G.A.M.); aekoshak@kau.edu.sa (A.E.K.); ssahmed@kau.edu.sa (S.S.E.); safadil@kau.edu.sa (S.A.F.); ali.abdullah.10@hotmail.com (A.A.A.); melfaky@kau.edu.sa (M.A.E.); 2Department of Pharmacognosy, Faculty of Pharmacy, Cairo University, Cairo 11562, Egypt; ali.elhalawany@pharma.cu.edu.eg; 3Department of Medicinal Chemistry, Faculty of Pharmacy, Suez Canal University, Ismailia 41522, Egypt; khaled_darwish@pharm.suez.edu.eg; 4Department of Biological Sciences, Faculty of Science, King Abdulaziz University, Jeddah 21589, Saudi Arabia; malgandaby@kau.edu.sa; 5Preparatory Year Program, Department of Chemistry, Batterjee Medical College, Jeddah 21442, Saudi Arabia; sabrin.ibrahim@bmc.edu.sa; 6Department of Pharmacognosy, Faculty of Pharmacy, Assiut University, Assiut 71526, Egypt; 7Department of Pharmacology and Toxicology, King Abdulaziz University, Jeddah 21589, Saudi Arabia; aaabdulalrahman1@kau.edu.sa

**Keywords:** SARS-CoV-2 main protease, coronavirus, SARS-CoV-2 Egyptian strain, thymoquinone, gardenin A, 6-gingerol, 6-paradol

## Abstract

Since the emergence of the pandemic of the coronavirus disease (COVID-19) caused by severe acute respiratory syndrome coronavirus 2 (SARS-CoV-2), the discovery of antiviral phytoconstituents from medicinal plants against SARS-CoV-2 has been comprehensively researched. In this study, thirty-three plants belonging to seventeen different families used traditionally in Saudi Arabia were tested in vitro for their ability to inhibit the SARS-CoV-2 main protease (M^PRO^). Major constituents of the bio-active extracts were isolated and tested for their inhibition potential against this enzyme; in addition, their antiviral activity against the SARS-CoV-2 Egyptian strain was assessed. Further, the thermodynamic stability of the best active compounds was studied through focused comparative insights for the active metabolites regarding ligand–target binding characteristics at the molecular level. Additionally, the obtained computational findings provided useful directions for future drug optimization and development. The results revealed that *Psiadia punctulata*, *Aframomum melegueta*, and *Nigella sativa* extracts showed a high percentage of inhibition of 66.4, 58.7, and 31.5%, against SARS-CoV-2 M^PRO^, respectively. The major isolated constituents of these plants were identified as gardenins A and B (from *P. punctulata*), 6-gingerol and 6-paradol (from *A. melegueta*), and thymoquinone (from *N. sativa*). These compounds are the first to be tested invitro against SARS-CoV-2 M^PRO^. Among the isolated compounds, only thymoquinone (THY), gardenin A (GDA), 6-gingerol (GNG), and 6-paradol (PAD) inhibited the SARS-CoV-2 M^PRO^ enzyme with inhibition percentages of 63.21, 73.80, 65.2, and 71.8%, respectively. In vitro assessment of SARS-CoV-2 (hCoV-19/Egypt/NRC-03/2020 (accession number on GSAID: EPI_ISL_430820) revealed a strong-to-low antiviral activity of the isolated compounds. THY showed relatively high cytotoxicity and was anti-SARS-CoV-2, while PAD demonstrated a cytotoxic effect on the tested VERO cells with a selectivity index of CC_50_/IC_50_ = 1.33 and CC_50_/IC_50_ = 0.6, respectively. Moreover, GNG had moderate activity at non-cytotoxic concentrations in vitro with a selectivity index of CC_50_/IC_50_ = 101.3/43.45 = 2.3. Meanwhile, GDA showed weak activity with a selectivity index of CC_50_/IC_50_ = 246.5/83.77 = 2.9. The thermodynamic stability of top-active compounds revealed preferential stability and SARS-CoV-2 M^PRO^ binding affinity for PAD through molecular-docking-coupled molecular dynamics simulation. The obtained results suggest the treating potential of these plants and/or their active metabolites for COVID-19. However, further in-vivo and clinical investigations are required to establish the potential preventive and treatment effectiveness of these plants and/or their bio-active compounds in COVID-19.

## 1. Introduction

In December 2019, in Wuhan, China, there was an ongoing epidemic of pneumonia relevant to the severe acute respiratory syndrome coronavirus 2 (SARS-CoV-2) [1]. There are six targets for SARS-CoV-2 that play a crucial role in the virus life cycle; the papain-like protease [2], the main protease (M^PRO^) [3], RNA-dependent RNA polymerase (RdRp) [4], non-structural protein (Nsp15) [5], spike protein [6], and angiotensin-converting enzyme (ACE2) [7].

The genomes of the SARS- and MERS-CoVs include two open reading frames, ORF1a and ORF1b, which are, respectively, translated by host ribosomes into two viral polyproteins, pp1a and pp1ab. Two cysteine proteases, a 3C-like protease (3CLpro) and a papain-like protease (PLpro), are encoded by ORF1a. While PLpro cleaves the polyprotein’s initial three cleavage sites, 3CLpro cleaves the other 11 sites, releasing a total of 16 non-structural proteins (nsp) for both SARS-CoV and MERS-CoV [8]. Targeting the main protease (3CL) as a primary enzyme mediated in the replication and transcription of the virus is a promising tool for the discovery of new antiviral drugs. The SARS-CoV-2 3CL (PDB ID: 6LU7) has a 96% resemblance to SARS-CoV 3CL. Since proteases are promising targets for SARS-CoV-2 replication inhibition, and amino acids Thr24, Thr26, and Asn119 3CL can play a part in binding with antiviral drugs, most of the medications currently available for the use of SARS-CoV-2 specifically work on the main protease (3CL) [9]. Natural products are a good source of bioactive antiviral compounds [10]. In this regard, diverse plants are proven to have a substantial traditional role in treating respiratory disorders and are a good source of bioactive antiviral metabolites. A number of plant-based natural compounds are under investigation, are in preclinical trials, or are in clinical trials. For example, (+)-Calanolide A from *Calophyllum langigerum* and SP-303 from the latex of *Croton lechleri* are under clinical investigation [11]. Moreover, different phyto-constituents were reported to be potent inhibitors against SARS-CoV-2, such as baicalin, ivermectin, and artemisinin. Their activity depends on the targeting of viral protease, virus entry, replication, and release from the infected cells [12]. It was also reported that thymoquinone, rosmarinic acid, ellagic acid, and thymol prevent virus entry. Meanwhile, quercetin and caffeic acid were reported as 3CL protease inhibitors [13]. To accelerate the discovery of antiviral hits against SARS-CoV-2, molecular docking for thousands of known phyto-constituents was performed on the main protease by utilizing computer-aided programs [14,15,16,17]. The compounds with high activity in virtual screening encourage researchers to test them in vitro on SARS-CoV-2 M^PRO^. These studies afforded many active compounds against the main protease, such as neoechinulin A from Red-Sea-derived *Aspergillus fumigatus* MR2012 [16], naringenin [17], and cnicin from *Carduus benedictus* [18]. Moreover, thirty-eight African medicinal plants traditionally used as an antiviral and immunomodulator and for treating COVID-19 symptoms were reviewed previously to open the door for drug discovery from medicinal plants [19].

In this work, in-house crude plant extracts were tested in vitro for their ability to inhibit the SARS-CoV-2 main protease (M^PRO^). Major constituents of the bio-active extracts were isolated and tested for their inhibitory capacity versus this enzyme as well as against the SARS-CoV-2 Egyptian strain. The most active compounds were further inspected by utilizing molecular docking and molecular dynamics to comprehend specific amino acids’ participation with the inhibitor at the active sites and to assess the thermodynamic stability.

## 2. Materials and Methods

### 2.1. Plant Material

Thirty-three plants belonging to seventeen different families were collected from Al-Taif and Al-Baha governorates between March and May 2020. *Aframomum melegueta* (AM-1307) was purchased from a local Saudi market. Identification of all plant samples was confirmed by members of plant taxonomy at College of Science, Jeddah University, Saudi Arabia. Voucher specimens were deposited at the herbarium of the Department of Natural Products and Alternative Medicine, College of Pharmacy, King Abdulaziz University, Jeddah, Saudi Arabia.

### 2.2. Preparation of the Crude Plant Extracts

Fifty grams of each dried plant material were extracted with methanol (3 × 200 mL) till exhaustion. The collected extracts were concentrated under vacuum and kept for biological study.

### 2.3. Isolation of Major Active Constituents

#### 2.3.1. Isolation of 6-Gingerol and 6-Paradol

Dried pulverized seeds (500 g) of *A. melegueta* (Roscoe) K. Schum (AM-1307) were extracted with MeOH until exhaustion. The total MeOH extract was evaporated under vacuum to give 30 g of dark brown residue that was suspended in the least amount of water and partitioned with chloroform (CHCl_3_); the pooled fractions were concentrated to yield 20 g dried extract. The chloroform fraction was chromatographed on SiO_2_ CC (silica gel column chromatography) and gradiently eluted with *n*-hexane-EtOAc (5–80% *v*/*v*) to obtain ten subfractions (1–10). Subfractions 3 and 7 contained two major spots. They were separately chromatographed on SiO_2_ CC using n-hexane-EtOAc (9:1 *v*/*v*) and n-hexane-EtOAc (7:3 *v*/*v*) to yield 6-paradol and 6-gingerol, respectively.

#### 2.3.2. Isolation of Gardenins A and B

Dried aerial parts of *P. punctulata* (500 g) (PP-1065) were extracted with methanol (2.5 L × 4). The obtained extract was evaporated to afford brown residue (50 g). This residue was suspended in water and partitioned with CHCl_3_. The collected fractions were concentrated to give 15 g of chloroform fraction that was then chromatographed on SiO_2_ CC using *n*-hexane-EtOAc gradient to obtain 4 major subfractions (1–4). Subfraction 1 contained two major spots that were submitted to SiO_2_ CC (30 g, 50 × 2 cm, n-hexane:EtOAc 95:5) to give gardenins A and B.

#### 2.3.3. Isolation of Thymoquinone

A total of 200 g *Nigella sativa* (NS-0801) dried pulverized seeds (250 G) was extracted with MeOH. The concentrated total extract (20 g) was suspended in water and partitioned with CHCl_3_. SiO_2_ CC of chloroform fraction (15 g) using n-hexane-EtOAc (97:3 *v*/*v*) afforded thymoquinone.

### 2.4. Identification of Isolated Compounds

The isolated compounds were identified utilizing spectroscopic data (e.g., ^1^H and ^13^C) or co-TLC in addition to comparison with the published data [20,21,22]. The spectral data of isolated compounds are represented in the Appendix A (Appendix A).

### 2.5. In Vitro Screening

M^PRO^ Protease, Untagged (SARS-CoV-2) Assay Kit, Catalog #: 78042-1, BPS Bioscience, Inc., San Diego, CA, USA), was used to investigate enzyme-inhibitory activities in vitro [23]. The inhibition was carried out using a fluorescent substrate containing the cleavage site (↓) of SARS-CoV-2 Mpro (Dabcyl-KTSAVLQ↓SGFRKM-E (Edans), M^PRO^ protease (SARS-CoV-2 M^PRO^ Protease), GenBank Accession No. YP 009725301, a.a. 1–306 (full length), expressed in *E. coli* expression system, MW 77.5 kDa, with a buffer containing 20 mM Tris, 100 mM NaCl, 1 mM EDTA, 1 mM DTT, pH 7.3, and GC376, an M^PRO^ protease inhibitor with a molecular weight of 507.5 Da. The fluorescence signal of the Edans generated due to the M^PRO^ Protease cleavage of the substrate was monitored using an Flx800 fluorescence spectrophotometer at an emission wavelength of 460 nm and an excitation wavelength of 360 nm in the FRET-based cleavage assay (BioTek, Winooski, VT, USA). Initially, 30 µL of diluted SARS-CoV-2 M^PRO^ protease was pipetted onto a 96-well plate containing 10 µL of pre-pipetted test compounds at a final concentration of 15 ng. The mixture was incubated for 30 min at room temperature (RT) with moderate shaking. The reaction was then started by adding the substrate (10 µL) dissolved in the reaction buffer to a final volume of 50 µL, at a concentration of 40 M, then incubated at RT for 4 h with gentle shaking. The plates were then sealed with tape. A microtiter plate-reading fluorimeter capable of excitation at 360 nm and detection of emission at 460 nm was used to assess fluorescence intensity.

### 2.6. MTT Cytotoxicity Assay

Half-maximal cytotoxic concentration (CC_50_) was assessed using 3-(4, 5-dimethylthiazol -2-yl)-2, 5-diphenyltetrazolium bromide (MTT) method in VERO-E6 cells, as previously described [17,24]. Tested compounds were prepared as a stock solution in 10% DMSO in double-distilled water, and final concentration was obtained by dilution with DMEM (Dulbecco’s Modified Eagle’s Medium).

### 2.7. Inhibitory Concentration 50 (IC50) Determination

IC_50_ of isolated compounds was assessed using hCoV-19/Egypt/NRC-03/2020 (accession number on GSAID: EPI_ISL_430820) that was adsorbed on Vero-E6 cells, as previously reported [25].

### 2.8. Molecular Modeling Investigation

Molecular docking was performed via MOE2019 software, as previously reported [26]. In brief, ligands were sketched and minimized through 0.0001 kcal/mol.Å^2^ gradient applying MMFF94s forcefields with standard ionization at pH 7.40. Crystalline structures of the SARS-Cov2 M^PRO^ proteins (PDB entries; 6W63 or 7CBT) were utilized for the non-covalent or covalent ligand docking protocols, respectively [27]. The PDB entry 7CBT was co-crystallized with the covalent inhibitor, GC376, which was used as positive control reference. However, the 6W63 PDB entry in complex with the non-covalent inhibitor, X77, was adopted for molecular docking of the isolated natural compounds against an M^PRO^ non-covalent reference inhibitor. Both proteins were prepared through standard preparation settings of 3D_protonation under physiological pH and Amber14:EHT forcefields. Covalent docking of GC376 at M^PRO^ was completed through acetalization reaction between the activated GC376 reactive aldehydic group and the catalytic Cys145 sulfhydryl group forming the hemi-thioacetal adduct [28]. Non-covalent docking within the M^PRO^ putative active site was completed through rigid docking protocol via Triangular-Matcher method, initially scored via London_dG, and final re-scoring with Generalized-Born solvation IV/Weighted-Surface Area dG throughout the post-placement refinement stage.

Adopting the rigid docking protocol was rationalized since several reported analyses of the M^PRO^ substrate-binding site considered it of limited flexibility (<2.00 Å RMSD) [29,30]. Additionally, aligning the M^PRO^ at apo (PDB: 6M03) and holo states with either non-covalent (PDB: 6W63 or 5R7Z) or covalent binders (PDB: 7JYC, 6XHO, or 7CBT) depicted great superimposition at the alpha-carbon RMSD at 0.395 Å, 0.377 Å, 0.288 Å, 0.519 Å, and 0.503 Å, respectively. The latter indicated a non-presentable difference between both M^PRO^ states with non-relevant impact of local-ligand-induced-fitting on M^PRO^-holo structures, at least within macromolecular crystallized states [31]. Validation of both docking protocols was performed through re-docking (self-docking) the co-crystallized ligands within the target’s canonical binding site [32]. Ligand’s pose prediction was completed by ranking the MOE-docking scores, RMSDs below a 2.00 Å threshold, and depicted relevant residue-wise binding interactions, as reported within literature. PyMol2.0.6 was used for pose visual inspection and binding interaction analyses [33].

GROMACS-2019 software package under CHARMM36m force field for protein and CHARMM-General Force Field program (Param-Chem project; https://cgenff.umaryland.edu/ accessed on 13 December 2021) for ligands was used to conduct the explicit molecular dynamics simulations [34,35]. Ligand–protein model was solvated within TIP3P cubic box under periodic boundary conditions with 10 Å marginal distances [36]. Protein residues were assigned at their standard ionization states (pH 7.4), while the entire system’s net charge was neutralized via potassium and chloride ions [34]. Constructed systems were minimized through 5 ps under the steepest descent algorithm [35], and they were then equilibrated for 100 ps under NVT ensemble (303.15 K) followed by 100 ps NPT ensemble (1 atm. pressure and 303.15 K) [37]. The production stage involved 100 ns MD simulation runs under NPT ensemble while using the Particle Mesh Ewald algorithm for computing the long-range electrostatic interactions [38]. Covalent bond lengths were modeled under LINCS with 2 fs integration time step size [39]. Both Coulomb’s and van der Waals’s non-bonded interactions were truncated at 10 Å using the Verlet cut-off scheme [40]. The binding-free energy between the ligand and protein, as well as the residue-wise contributions within the binding-free energy calculations, were estimated via MM/PBSA on representative frames for the whole-MD simulation runs (100 ns) [41].

### 2.9. Statistical Analysis

IC_50_s were estimated by Graph-pad-Prism 8^®^. One-way ANOVA followed by Tukey’s test was utilized for calculation of significant differences between means.

## 3. Results and Discussion

### 3.1. SARS-CoV-2 MPRO Inhibitory Activity of Plant Extracts and Major Isolated Constituents

Thirty-three plant extracts were screened for their inhibitory activity against SARS-CoV-2 M^PRO^ using a FRET assay at 100 ug/mL, and GC376 was used as a positive control. The results (Table 1, Figure 1) reveal the activity of *Psiadia punctulata*, *Echinops macrochaetus*, *Abutilon pannosum*, *Lavandula dentata*, *Cometes abyssinica*, *Aframomum melegueta*, and N*igella* sativa extracts with percentage inhibitions of 66.4, 8.13, 3.2, 4.43, 0.83, 58.7, and 31.5%, respectively.

The potent extracts with significant inhibition of the viral protease (Figure 1), including *P. punctulate*, *A. melegueta*, and *N. sativa*, were subjected to different chromatographic procedures to isolate their major constituents.

Gardenins A and B were isolated from *P*. *punctulata.* Meanwhile, 6-gingerol and 6-paradol were isolated from *A*. *melegueta,* and thymoquinone was isolated from *N*. *sativa* seeds. The isolated compounds were also tested for their inhibitory activity against SARS-CoV-2 M^PRO^ using a FRET assay.

The results (Table 2) reveal the ability of thymoquinone, gardenin A, 6-gingerol, and 6-paradol to inhibit the SARS-CoV-2 M^PRO^ enzyme with inhibition percentages of 63.21, 73.80, 65.2, and 71.8%, respectively.

Moreover, their IC_50_ was assessed relative to GC376 (Standard M^PRO^ Protease enzyme inhibitor) on the SARS-COV-2 viral main protease (Table 3 and Figure 2). 6-paradol showed the highest potency (IC_50_ 0.1682 µM), followed by gardenin A, 6-gingerol, and thymoquinone, with an IC_50_ of 5.964, 9.327, and 10.26µM, respectively.

### 3.2. Antiviral Activity of Isolated Compounds on the SARS-CoV-2 Viral Main Protease Inhibitors

Based on the obtained IC_50_ (Table 3), the active compounds (thymoquinone, 6-gingerol, 6-paradol, and gardenin A) were tested for their anti-viral activity on SARS-CoV-2 (hCoV-19/Egypt/NRC-03/2020 (accession number on GSAID: EPI_ISL_430820). The results (Figure 3) indicated that tested compounds displayed strong-to-low (5 µg/mL < IC_50_ < 100 µg/mL) antiviral activity against SARS-CoV-2.

Major constituents of *Nigella sativa* were screened virtually against 3CLpro (the main protease) and NSP15 (nonstructural protein 15 or exonuclease). The results showed that dithymoquinone has promising binding activity against these two targets, which suggests this molecule is a potential therapeutic molecule against COVID-19 [42]. Thymoquinone (THY) is one of the major constituents (28–57%) of black seed (*N. sativa*) oil. It has different biological activities, including cytotoxic, anti-inflammatory, anti-microbial, anti-viral, anti-oxidant, immune-stimulant, and anticoagulant effects [43]. Moreover, it has the ability to reduce the levels of pro-inflammatory mediators (ILs 2, 4, 6, and 12) and increases INF-γ [44]. THY is able to increase IgG1 and IgG2a levels as well as improve pulmonary function tests in restrictive respiratory disorders [44]. Previous in silico studies reported the ability of THY to inhibit the SARS-CoV-2 protease [45] as well as ACE2 [46].

Moreover, it has the ability to block the binding of the viral S-protein to the cellular receptor ACE2 of designed coronavirus pseudoparticles, thus blocking viral entry into the host cell. Our results proved, for the first time, the ability of THY to inhibit the SARS-CoV-2 viral main protease in vitro with IC_50_ 10.26 µM. THY showed relatively high cytotoxicity and strong anti-SARS-CoV-2 activity with a low selectivity index (CC_50_/IC_50_ = 8.2/6.14 = 1.33).

From a clinical point of view, THY was safe at a dose of 5 mg/day to 2600 mg/day without any toxic symptoms [47]. Pharmacokinetic studies were performed after oral and IV administration, and they showed that THY has rapid-elimination properties and relatively slower absorption following oral administration [48].

6-Gingerol (GNG) and 6-paradol (PAD) are the major constituents of *Aframomum melegueta* seeds. The seed is traditionally used for respiratory tract infections, tuberculosis, and coughs [19]. Previous reports proved the anti-inflammatory and immunomodulatory properties of the seed [19]. Previous in silico studies and molecular dynamics showed that GNG has binding affinities of −5.60, −5.40, and −5.37 (kcal/mol) against Cathepsin K, the COVID-19 main protease, and the SARS-CoV 3 C-like protease, respectively [49]. Meanwhile, it showed low potency against SARS-CoV-2 infection with IC_50_ > 100 μM (CC_50_ > 100 μM) [50]. In this study, for the first time, GNG was tested on the SARS-CoV-2 viral main protease in vitro with IC_50_ 9.327 µM. Although previous results showed a low potency of GNG on the virus [50], our data showed moderate activity against the SARS-CoV-2 virus at non-cytotoxic concentrations in vitro with a significant selectivity index (CC_50_/IC_50_ = 101.3/43.45 = 2.3).

Meanwhile, PAD was not tested before, in silico or in vitro, against the six targets for SARS-CoV-2. Although the obtained results showed the highest potency (IC_50_ 0.1682 µM) against the SARS-CoV-2 viral main protease, it showed a cytotoxic effect on the tested VERO cells with a selectivity index of CC_50_/IC_50_ = 155.1/281 = 0.6, which indicated the unsuitability of this method to detect its activity.

Pharmacokinetic studies proved low levels of toxicity and high levels of tolerability of GNG in humans at doses up to 2.0 g daily, with only mild gastrointestinal complaints being reported [51]. GNG is well-absorbed after oral administration and detected in blood as glucuronide and sulfate conjugates [49,51].

Gardenin A (GDA) was not tested before for its activity on SARS-CoV-2. It showed week activity on SARS-CoV-2 virus at non-cytotoxic concentrations in vitro with a significant selectivity index (CC_50_/IC_50_ = 246.5/83.77 = 2.9).

### 3.3. Molecular Modeling of the In-Vitro-Active Compounds

An initial molecular docking study was conducted to investigate the differential binding poses of the in-vitro-active metabolites towards the SARS-Cov2 M^PRO^ active site. The computational study was also beneficial for identifying the key residues involved in ligand–target binding interactions. The above in vitro reference inhibitor, GC376, in complex with M^PRO^ (PDB: 7CBT, reported bioassay IC_50_ = 0.026 μM), was also adopted for the presented in silico study as a positive control. Typically, GC376 is a bispeptidyl disulfite adduct salt that is well-recognized as a broad-spectrum prodrug exerting strong inhibitory activities against coronaviruses and picornaviruses [52,53,54]. However, this positive control inhibitor has been reported to exert its M^PRO^ inhibition through covalent interaction within the target catalytic site [28,55]. Therefore it was highly rationalized to utilize another crystallized target (PDB: 6W63) bounded to a non-covalent M^PRO^ inhibitor, X77 (reported bioassay IC_50_ = 2.3 μM), as a second positive control to mimic the predicted binding interactions of our isolated plant metabolites. The latter non-proteinomimetic M^PRO^ inhibitor is an imidazole-carboxamide-based small molecule exhibiting potent broad-spectrum activity against coronaviruses [27]. In this regard, investigating the docking binding energies (kCal/mol) and M^PRO^ residue-wise binding interactions of the isolated compounds in relation to both covalent and non-covalent reference ligands highlight the key structural activity features required for M^PRO^ inhibition.

Throughout the molecular docking protocols, docked ligands showed relevant anchoring at the M^PRO^ binding site (Figure 4A). The SARS-CoV-2 protease is of similar topology as any M^PRO^ protease enzyme, where the substrate-binding site comprises four important subsites, S1′, S2, S3, and S4, correlating to the peptide-based substrate residues (P1′, P2, P3, and P4, respective) [56]. Both 6-paradol (PAD) and 6-gingerol (GNG) depicted common conformation/orientation, with their substituted aromatic scaffold being settled at the S1 subsite while their tail was extended across the other subsites, at the end reaching towards the S3 subsite. The flavonoid-based ligand, gardenin A (GDA), predicted extended and almost linear conformation across several M^PRO^ subsites due to its inherited structural rigidity related to its chromone ring. This ligand showed the deepest anchoring towards the S3 subsite via its tris-methoxy-substituted phenyl ring. The small-sized monoterpene molecule, thymoquinone (THY), exhibited significant orientation at the M^PRO^ pocket, being almost limited to the S1′ and S1 subsites. Limited orientation to M^pro^-pocket was also reported for small molecules docked at the M^pro^ pocket, which was correlated with their modest M^pro^-binding affinities [57]. Both reference inhibitors illustrated significant accommodation of the M^PRO^ pocket, with X77 being the one depicting the most extended conformation, reaching up to all four M^PRO^ subsites via its four ring/alkyl scaffolds. On the other hand, GC376 showed an enclosed conformation, with its two terminal rings deeply anchored at the S1 subsite, while its peptidomimetic linker adopted a C-shaped twist to allow significant contact with other subsites. Nevertheless, the anchoring of GC376 was more limited at S2 and S3 subsites compared to the other M^PRO^ subsites. Validation of the obtained binding modes was confirmed through self-docking (re-docking) studies where small root-mean-square deviations (RMSDs) were depicted for the positive controls in relation to their co-crystallized heavy atoms (0.920 Å and 1.779 Å for X77 and GC376, respectively) (Appendix A, Appendix A). The latter ensured the validity and accuracy of the adopted docking protocols and algorithms in predicting the best ligand binding mode while ensuring the biological significance of these furnished poses and their respective energies [58].

The anchoring of docked ligands within the M^PRO^ pocket was mediated through binding interactions with several key pocket residues (Table 4). Only PAD managed to depict relevant hydrogen bonding with both S1′ catalytic dyads, His 41 and Cys145, via the ligand’s carbonyl group at its aliphatic tail (Figure 4B). The latter suggested significant PAD inhibition activity of the M^PRO^ enzymatic machinery. It has been reported that these residues correspond to strong ligand–target binding where several natural M^pro^-potent inhibitors are reported, such as residue-wise binding [35,59,60,61]. Further stability of PAD at M^PRO^ was mediated through polar contacts with the Asn142 sidechain and Phe140 mainchain via the ligand’s oxygen functionalities on its aromatic scaffold. Although GNG shared close structural similarity to PAD, the earlier ligand failed to exhibit polar contacts with the catalytic residue Cys145 sidechain while keeping its aromatic head deeply anchored at the S1 subsite. Hydrogen bond pairing for the GNG’s free hydroxyl groups with His41 and His163 sidechains was shown instead. Depicting these binding interactions could be due to the presence of extra OH at the GNG’s aliphatic linker. The latter extra functionality might impose steric hindrances on surrounding residues as well as polar attraction forces better than the linker carbonyl, causing the latter to be at the solvent side while forcing the GNG’s head deeper into the S1 subsite. The latter differential binding mode was translated into higher docking binding energies for PAD over GNG (−6.3012 vs. −5.8539 kCal/mol). Regarding the flavonoid ligand, GDA, only two hydrogen bindings were depicted for the chromone oxygen functionalities towards the His41 and Ser144 mainchains. Due to the ligand’s limited maneuvers, GDA showed no significant polar contacts with the hydrophilic residues lining the S3 subsite; this could be correlated to a docking binding energy (−5.5162 kCal/mol) comparable to GNG.

Limited docking of THY at S1′ and S1 subsites with only two polar interactions with Gly143 and Glu166 mainchains was reasonably translated to the ligand’s lowest docking energy (−4.2899 kCal/mol) among all docked ligands. Moving towards the positive control ligands, both X77 and GC376 exhibited the most extended polar contacts with several pocket residues, including His164, Cys145, Gly143, His163, Phe140, and/or Glu166, comprising a large number of M^PRO^ subsites. The latter was translated to high docking energies of −8.5790 and −9.0396 kCal/mol for X77 and GC376, respectively. Notably, GC376 exhibited double-bonding with the catalytic Cys145, where the ligand furnished the hemi-thioacetal adduct with the residue’s SH group while depicting polar hydrogen bonding with its NH mainchain. The above docking highlights the key role of the Glu166 mainchain NH group in stabilizing both drug-like molecules (THY/X77) and the proteinomimetic ligand (GC376) at the M^PRO^ pocket. The highlighted Glu166 role came in great agreement with several reported studies [3,30,57,59,62].

The contribution of hydrophobic interactions within the depicted docking binding scores was further evaluated through an investigation of ligand-M^PRO^ van der Waals and π-mediated interactions. Interestingly, the top-docked ligands showed a wider range of non-polar contacts with several hydrophobic residues, including Met49, Phe140, Leu141, His163, His164, Met165, and/or Leu167. Several reports illustrated the significant role of the latter hydrophobic contacts, where His41, Met49, and Asp187 at S2 subsite as well as Met165 and Gln189 from subsite S3 served as the hydrophobic grip for pinning ligands at the M^PRO^ target pocket [3,59,62,63]. Furthermore, almost all ligands except THY showed a significant van der Waals interaction with the carbon sidechains of lining residues (Glu166, Asp187, and Gln189), and this was more extended for PAD and both positive controls. Despite being polar or even charged at physiological pH, the latter residues could manage to achieve non-polar contacts via their sidechain C*β* and/or C*δ* atoms, as previously reported in the current literature [30,63]. Finally, ligand stability was further mediated through π-driven non-polar interactions for the PAD, THY, and X77 molecules towards Phe140, His163, and/or His164 sidechains. Despite incorporating aromatic scaffolds within their molecular structures, both GNG and GC376 failed to depict relevant π-interactions due to unfavorable proximity/orientation towards the aromatic lining residues.

A molecular dynamic simulation was further performed to gain more insights regarding the differential thermodynamic behavior of the investigated ligand-M^PRO^ complexes at near-physiological conditions. Moreover, molecular dynamics studies would provide a scientific-based approach to validate the creditability of the obtained molecular docking binding interactions [64]. The stability of the simulated ligand-M^PRO^ complexes was illustrated by monitoring the RMSD trajectories of the combined ligand–target complex across the whole 100 ns explicit molecular dynamics simulation runs (Figure 5). Typically, RMSD measures the molecular deviation from the initial reference structure and provides a good indication of molecular stability and simulation validity. High target RMSDs confer instability and significant conformational changes [65] and correlate with weak ligand/target affinity, being incapable of accommodating the ligands within the protein’s pocket across the simulation timeframe [66].

Herein, steady RMSD tones were illustrated for PAD and X77-bound target complexes, reaching their respective dynamic equilibration plateaus after the initial 20 ns until the end of the simulation run. These depicted RMSDs were around average values of 2.58 ± 0.32 Å and 3.02 ± 0.23 Å for X77 and PAD, respectively, and were maintained for more than half of the molecular dynamic simulation runs (>75 ns). This thermodynamic behavior indicated the significant convergence and stability of these bounded M^PRO^ proteins as well as the adequacy of minimization/equilibration stages prior to the simulation production run, requiring no further simulation time extensions. These convergence and stability findings were consistent with reported studies investigating small molecule’s affinity to the SARS-CoV-2 M^pro^ target through molecular docking and dynamics simulations. The depicted PAD and X77-related RMSD tones were as steady as those for investigated top-active anti-M^pro^ beta-blocker agents, angiotensin II receptor blockers, marine polyketides, scalarane sesterterpenes, and natural flavonoid aglycones [60,61,67,68,69]. The latter confirmed the thermodynamic stability of PAD at the M^pro^ pocket.

Regarding GNG- and GDA-bound complexes, the RMSDs were steady for the first half of the simulation run, and then high fluctuations were depicted, conferring great ligand-pocket instability and poor accommodation within the active site. The highest RMSD trajectory fluctuations were assigned to the monoterpene THY complex, where at early frames, the RMSDs spiked high up to ~6.00 Å and continued until the end of the simulation run. Higher RMSD fluctuations were reported as significant for the conformational shift during dynamic simulations as well as non-confinement for poor M^pro^-binding ligands, as seen in studies by Al-Karmalawy and his research groups [70,71]. Moving towards the covalently bound GC376 complex, steady low RMSD trajectories (2.70 ± 0.21 Å) were depicted until 50 ns, where the tones slightly rose to ~4.00 Å, where they showed limited fluctuations until the end of the simulation run. The latter dynamic behavior highlighted a significant conformation shift for the GC376 ligand but with the retainment of the ligand at the M^PRO^-active site.

The above differential ligand-M^PRO^ complex stabilities were confirmed through conformational analysis of the simulated complexes at the initial and final timelines of the simulated runs (Figure 6). Extracted and 0.00001 kcal/mol.Å^2^ gradient-minimized frames at 0 ns and 100 ns only showed ligand-pocket confinement for PAD-, X77-, and GC376-bound M^PRO^ complexes. The latter came in great concordance with the above-obtained RMSD trajectories for the latter three top-stable complexes. Limited conformation alterations were depicted for X77, which corresponded to the lowest RMSD values across all simulated models. Furthermore, a significant conformational shift was illustrated for the scaffold of the GC376, where at the end of the simulation run, this functional group was shifted away from the pocket contact towards the solvent side. This could be the reason why there was a slight elevation at the RMSDs tones beyond the 60 ns timeframes. Regarding the other three simulated complexes, the bound ligands drifted away towards the solvent side for both THY and GDA at the end of their respective simulation runs. However, GNG rested at a surface cleft ~25 Å far from the canonical M^PRO^ binding site when it reached the final simulation frame. The latter could describe the steady RMSD tones depicted for GNG across 90–100 ns timeframes (Figure 5).

Exploring the nature of ligand-M^PRO^ binding as well as the individual ligand contributions was finally completed by calculating the free binding energies for each simulated model using the Molecular Mechanics/Poisson–Boltzmann Surface Area (MM/PBSA) calculation [72]. While a greater negative binding energy correlates to a higher ligand–target affinity, the MM/PBSA binding energy calculation accounts for more accurate ligand–target affinity compared to static or even most sophisticated flexible molecular docking techniques. The MM/PBSA approach is considered of comparable accuracy to Free-Energy Perturbation approaches, but at much less computational expenditure [41]. To our delight, significant free binding energies were depicted for the top-stable isolated compound, PAD, in relation to the reference potent M^PRO^ inhibitors (Table 5). Slightly high negative free binding energy was assigned to the covalent ligand over the non-covalent, which was higher than that of PAD. Moderate and comparable free binding energies were assigned to GDA and GNG, correlating to their weak affinity towards the M^PRO^ pocket. A poor energy value was depicted for THY, which came in great agreement with the preliminary docking scores as well as the above-described RMSD and conformational analyses.

Dissecting the total free binding energy in terms of its constituting energy terms illustrated dominant energy contributions of the van der Waal potentials over the Coulomb’s electrostatic attraction forces. This came with the cited reports that the M^PRO^ pocket is considered a large surface area that is mainly hydrophobic in nature [3,30,57,59,62]. Due to differential energy contributions of the closely related molecules, PAD and GNG, it was noticed that higher electrostatic energy was assigned for GNG, which could be correlated with the existence of an extra free polar hydroxyl group at the GNG linker. However, this could be double-bladed since GNG was assigned a much higher polar solvation energy (almost two-fold) compared to that of PAD, which might contribute to the earlier compromised stability and solvent drift. This is highly plausible since ligand–target binding is considered a solvent-displacement process. Similar findings were also depicted for the second unstable ligand, GDA, where the presence of a high number of polar oxygen functionalities on the flavonoid scaffold contributed to both high electrostatic and unfavored polar solvation energies, with the latter compromising the ligand affinity. Despite the high polar-solvation energies of the two control inhibitors, as they harbor several hydrogen bond donors/acceptors, both ligands managed to compensate for these repulsive forces due to their aromatic and hydrophobic functionalities. This was obvious by their furnishing of the highest high van der Waal energy contributions.

The depicted double-bladed influence of polar functionalities on SARS-CoV-2 M^pro^ ligand binding came in agreement with reported studies. Investigated scalarane sesterterpene metabolites isolated from *Hyrtios erectus* marine sponge were of limited solvation penalty towards M^pro^ binding compared to the more hydrophilic reference control, lopinavir [63]. The authors suggested that possessing the hydrophobic cage-like sesterterpene skeleton was beneficial for limiting the solvation penalty against their preferential binding by furnishing balanced compensatory van der Waal binding potentials. Similarly, Hassan and his research group reported higher solvation energies for investigated marine natural polyketides through molecular-docking-coupled dynamics investigations [60]. The hydrophilic functionalities decorating the ligands terminal tail were suggested to impose high solvation entropy against their M^pro^ binding. The authors suggested future optimization of the isolated compounds through the introduction of ionizable groups with higher hydrophobic characteristics (i.e., a tetrazole scaffold). This would have been beneficial for minimizing the solvation entropy and extending ligand/M^pro^ binding. In this regard, the ability of the ligand to exhibit balanced hydrophobic and hydrophilic potentialities compensating for the unfavored polar solvation repulsion forces would guarantee high affinity towards the M^PRO^-active site.

## 4. Conclusions

The emergence of SARS-CoV-2 caused more than 6 million deaths, and it also had major economic consequences on various countries of the world. Further, SARS-CoV-2 virus mutation increased the virus’s capacity to spread and infect. Despite the emergence of many vaccines, the lack of an effective treatment so far may exacerbate the problem. In this scenario, broad-spectrum and safe anti-SARS-Cov-2 treatments are urgently required. Researchers in various countries of the world are struggling to find a treatment for this virus and are exploring different methods. Molecular docking is used to accelerate the discovery of antiviral hits against SARS-CoV-2 by studying the effect of known phyto-constituents as well as known antiviral drugs on different targets of the virus that play an important role in its life. In this regard, medicinal plants could be a valuable source of simple bioactive compounds. In this work, thirty-three plants belonging to seventeen different families were tested against SARS-CoV-2 M^PRO^ using a FRET assay and GC376 as a positive control. *P. punctulata*, *A. melegueta*, and *N. sativa* extracts showed a high percentage of inhibition. Their isolated major constituents are gardenins A and B (from *P. punctulata*), 6-gingerol and 6-paradol (from *A. melegueta*), and thymoquinone (from *N. sativa*). Among them, only THY, GDA, GNG, and PAD showed SARS-CoV-2 M^PRO^ inhibition potential. It is worth noting that these compounds were tested for the first time in vitro on SARS-CoV-2 M^PRO^. Moreover, the tested compounds displayed moderate-to-low antiviral activity against SARS-CoV-2 ((hCoV-19/Egypt/NRC-03/2020). PDA showed the highest potency against the enzyme, but it demonstrated a cytotoxic effect on the tested VERO cells, which indicated the unsuitability of this method to detect its activity. THY showed relatively high cytotoxicity and strong anti-SARS-CoV-2 activity with a low selectivity index. Future studies on this compound via the preparation of different derivatives may help in increasing the selectivity index. Meanwhile, GNG had moderate in vitro activity at non-cytotoxic concentrations, with a significant selectivity index. Moreover, GDA possessed weak activity against the SARS-CoV-2 virus at non-cytotoxic concentrations in vitro.

An in silico study replicated the depicted in vitro findings and showed the stability of several isolated metabolites, particularly PAD, towards the SARS-CoV-2 M^PRO^ main enzyme. These findings introduced PAD as a promising clinical candidate for further drug optimization and development processes.

## Figures and Tables

**Figure 1 plants-11-01914-f001:**
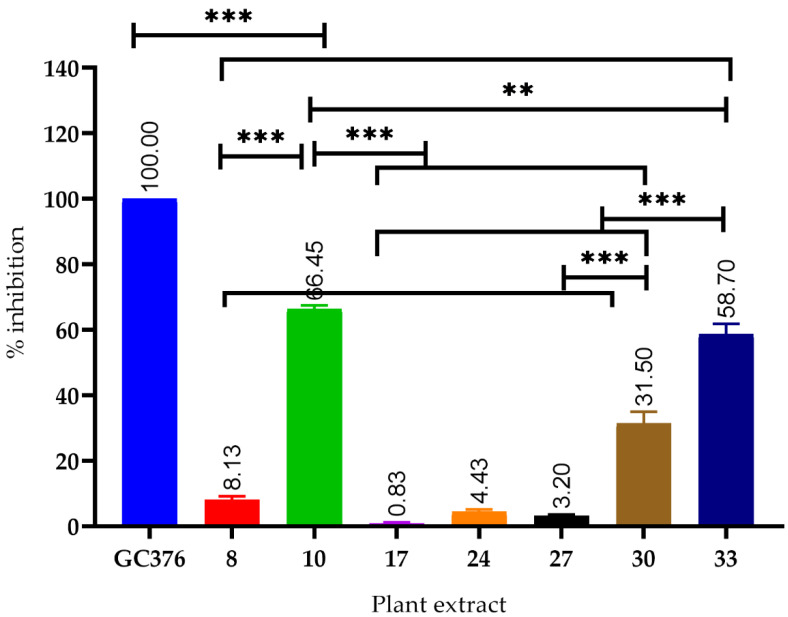
Plant extracts with significant inhibitory activity against the viral protease (SARS-CoV-2 M^PRO^) GC376; positive control, *** significantly different at *p* < 0.0001, ** significantly different at *p* < 0.001.

**Figure 2 plants-11-01914-f002:**
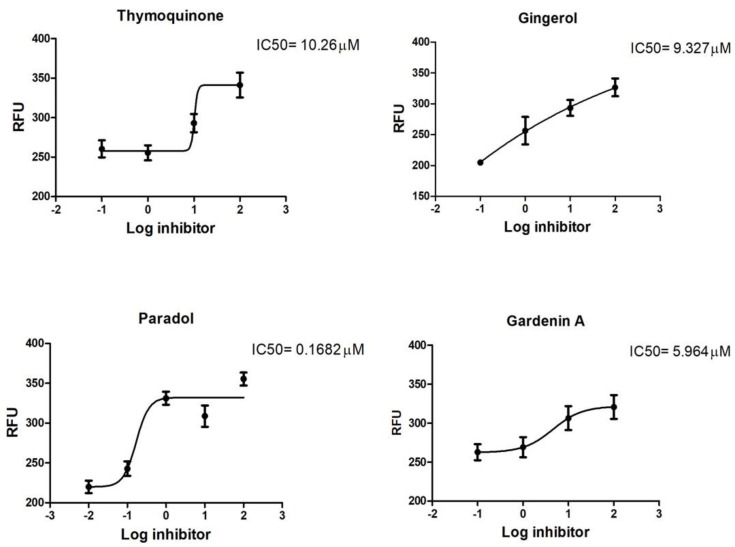
Inhibition of M^PRO^ protease enzyme activity by isolated compounds.

**Figure 3 plants-11-01914-f003:**
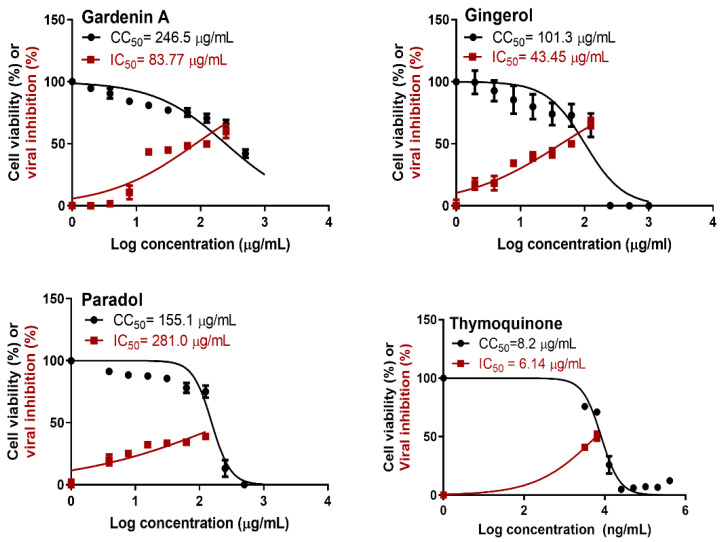
Dose-inhibition curves for bioactive compounds, where IC50 values were calculated using nonlinear regression analysis of GraphPad Prism software (version 5.01) by plotting log inhibitor versus normalized response.

**Figure 4 plants-11-01914-f004:**
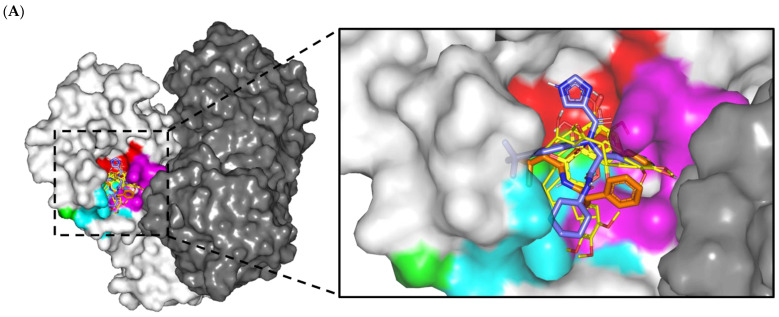
Ligand/M^PRO^ binding modes and interactions. (**A**) Surface rendition of SARS-CoV-2 M^PRO^, with an overlay of docked isolated compounds (yellow lines) and potent reference (blue or orange sticks for X77 or GC376, respectively). The protein is colored in dark and light gray colors for protomer A and B, respectively, while the target binding subsites are shown in red, magenta, green, and cyan for S1′, S1, S2, and S3 subsites, respectively; (**B**) Docked binding modes of investigated compounds (sticks), where residues (lines) only located within 5Å radius of bound ligands are displayed, labeled by sequence numbers, and colored based on respective subsite location. Polar interactions (hydrogen bonds) are shown as dashed black lines.

**Figure 5 plants-11-01914-f005:**
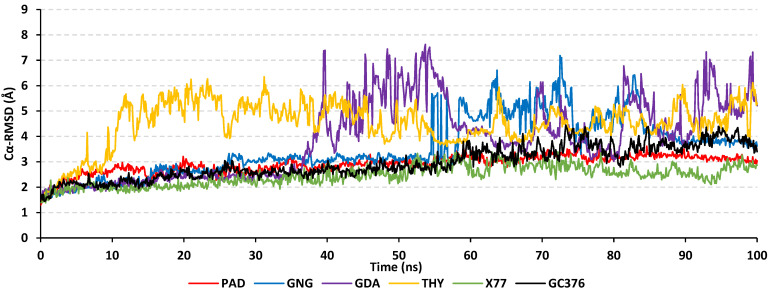
Stability analysis of the ligand-M^PRO^ complex across the 100 ns explicit molecular dynamics simulation runs. The generated RMSD trajectories (Å) are represented across the simulation timeframes (ns).

**Figure 6 plants-11-01914-f006:**
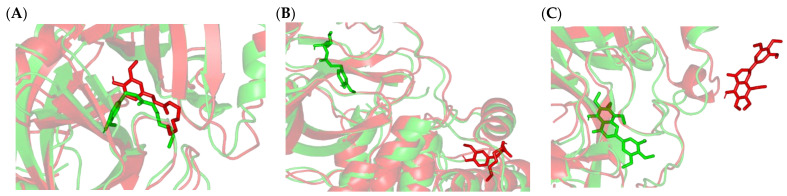
Conformational analysis of the simulated ligand-M^PRO^ complexes at the start and end of the 100 ns explicit molecular dynamics simulation runs. (**A**) PAD; (**B**) GNG; (**C**) GDA; (**D**) THY; (**E**) X77; (**F**) GC376. Overlaid snapshots at 0 ns and 100 ns are presented in green and red colors, respectively; the target proteins (cartoon) and ligands (sticks) are colored corresponding to the extracted frame.

**Table 1 plants-11-01914-t001:** Inhibitory activity of the tested plant extracts against SARS-CoV-2 viral main protease (SARS-CoV-2 M^PRO^).

Family	Sample No.	Plant Name	Specimen Number	% Inhibition
Acanthaceae	1	*Barleria trispinosa* (Forssk.) Vahl	BT-1003	-
Amaranthaceae	2	*Traganum nudatum* Delile	TN-0554	-
Apocynaceae	3	*Caralluma russelliana* (Courbai ex Brongn.) Cufod.	CR-1168	-
	4	*Leptadenia pyrotechnica* (Forssk.) Decne.	LP-0840	-
	5	*Rhazya stricta* Decne.	RS-1014	-
Asparagaceae	6	*Dracaena cinnabari* Balf.f. Resin	DC-1140	-
Asteraceae	7	*Conyza pyrrhopappa* Sch.Bip. ex A.Rich.	CP-1060	-
	8	*Echinops macrochaetus* Fresen.	EM-0535	8.13
	9	*Lactuca serriola* L.	LS-1050	-
	10	*Psiadia punctulata* Vatke	PP-1065	66.45
	11	*Pulicaria arabica* (L.) Cass.	PA-0733	-
	12	*Saussurea lappa* (Decne.) Sch.Bip	SL-1050	-
	13	*Tagetes minuta* L.	TM-3015	-
	14	*Verbesina encelioides* (Cav.) Benth. & Hook. f. ex A.Gray	VE-1053	-
Brassicaceae	15	*Lepidium sativum* L.	LS-1040	-
Capparaceae	16	*Maerua crassifolia* Forssk.	MC-1034	-
Caryophyllaceae	17	*Cometes abyssinica* R. Br. Ex Wall.	CA-1315	0.83
Cleomaceae	18	*Cleome viscosa* L.	CV-0541	-
Convolvulaceae	19	*Convolvulus glomeratus* Choisy.	CG-0440	-
Fabaceae	20	*Crotalaria emarginella* Vatke	CE-1119	-
	21	*Pithecellobium dulce* (Roxb.) Benth.	PD-1020	-
	22	*Parkinsonia aculeata* L.	PA-1166	-
	23	*Tephrosia nubica* (Bioss.) Baker	TN-1120	-
Lamiaceae	24	*Lavandula dentata* L	LD-1090	4.43
	25	*Origanum majorana* L.	OM-1239	-
	26	*Phlomis floccosa* D.Don	PF-1140	-
Malvaceae	27	*Abutilon pannosum* (G.Forst.) Schltdl.	AP- 1041	3.2
	28	*Triumfetta flavescens* Hochst. ex A.Rich.	TF-1142	-
Peraceae	29	*Clutia myricoides* Jaub. & Spach	CM-1088	-
Ranunculaceae	30	*Nigella sativa* L. seeds	NS-0801	31.5
Solanaceae	31	*Solanum surattense* Burm. f.	SS-1141	-
	32	*Withania somnifera* (L.) Dunal	WS-1154	-
Zingiberaceae	33	*Aframomum melegueta* K. Schum	AM-1307	58.7

**Table 2 plants-11-01914-t002:** Percentage inhibition of the isolated compounds from bioactive extracts against viral protease (SARS-CoV-2 M^PRO^).

Compound Name	Structure	% Inhibition
Thymoquinone	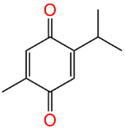	63.21
Gardenin B	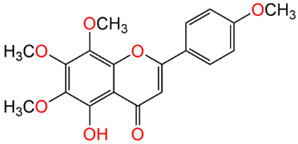	1.00
Gardenin A	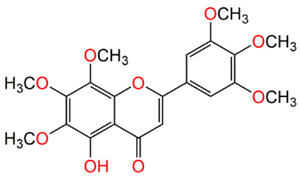	73.80
6-Gingerol	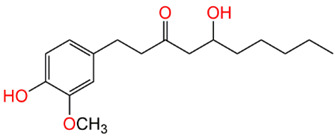	65.2%
6-Paradol	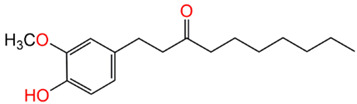	71.8%

**Table 3 plants-11-01914-t003:** IC_50_ of isolated compounds compared to GC376 (Standard M^PRO^ protease enzyme inhibitor) on SARS-CoV-2 viral main protease.

Compound Name	IC_50_ (µM)
Thymoquinone	10.26
Gardenin A	5.964
6-Gingerol	9.327
6-Paradol	0.1682
GC376 (positive control)	0.0012

**Table 4 plants-11-01914-t004:** Ligand-target binding interaction parameters for the docked ligands and positive controls within M^PRO^ binding site.

Compound	Docking Energy (Kcal/mol)	H-Bond Interactions (Distance Å; Angle °)	Hydrophobic Interactions/van der Waal with Side Chain Carbons	π-Interaction (Distance Å)
PAD	−6.3012	His41 (3.2 Å; 121.6°), Cys145 (3.4 Å; 125.7°), Phe140 (1.9 Å; 129.5°), Asn142 (1.4 Å; 149.7°)	Met49, His164, Met165, Leu167/Asp187 (C*β*,C*δ*), Gln189 (C*β*,C*δ*)	Phe140 (π-H) (3.7 Å)
GNG	−5.8539	His41 (3.4 Å; 120.4°), His163 (3.0 Å; 161.0°)	Met49, Phe140, Met165, His164/Asp187 (C*β*), Gln189 (C*β*)	—
GDA	−5.5162	His41 (3.1 Å; 123.9°), Ser144 (2.6 Å; 141.1°)	Met165, Leu167, His164/Glu166 (C*β*,C*δ*)	Phe140 (π-H) (5.2 Å) His163 (π-π) (5.3 Å)
THY	−4.2899	Gly143 (2.4 Å; 164.4°), Glu166 (1.8 Å; 132.3°)	Phe140, Met165, His164	His163 (π-H) (4.5 Å)
X77	−8.5790	Gly143 (2.4 Å; 145.6°), Gly143 (2.7 Å; 139.5°), His163 (3.0 Å; 142.7°), Glu166 (2.1 Å; 161.9°)	Met49, Phe140, Leu141, His163, Met165, Leu167/ Asn142 (C*β*,C*δ*), Asp187 (C*β*,C*δ*), Gln189 (C*β*,C*δ*)	His164 (π-π) (5.1 Å)
GC376	−9.0396	Phe140 (1.7 Å; 139.1°) Gly143 (2.5 Å; 138.3°), Cys145 (2.3 Å; 158.3°), His164 (2.3 Å; 169.6°), His163 (2.6 Å; 130.6°), Glu166 (2.6 Å; 130.6°)	Met49, Phe140, Leu141, His163, Met165, Leu167/ Asn142 (C*β*,C*δ*), Asp187 (C*δ*), Glu166 (C*β*,C*δ*)	—

**Table 5 plants-11-01914-t005:** Binding-free energies and dissected contribution energy terms of the ligand-M^PRO^ complexes.

Energy (kJ/mol ± SE)	Ligand-M^PRO^ Complexes					
	PAD	GNG	GDA	THY	X77	GC376
van der Waal	−200.525 ± 14.859	−208.826 ± 36.878	−297.374 ± 20.396	−81.297 ± 80.752	−290.680 ± 14.694	−338.201 ± 19.508
Electrostatic	−55.142 ± 6.607	−88.152 ± 21.814	−96.964 ± 5.328	−32.173 ± 31.394	−127.985 ± 6.912	−130.781 ± 3.844
Solvation; Polar	145.170 ± 45.751	228.655 ± 30.934	318.944 ± 16.085	108.316 ± 92.295	274.061 ± 9.408	340.822 ± 20.096
Solvation; Apolar-SASA	−29.505 ± 2.352	−33.672 ± 1.436	−36.409 ± 2.272	−9.327 ± 12.288	−20.797 ± 1.356	−40.481 ± 0.628
Total binding energy	−140.002 ± 24.642	−101.995 ± 30.019	−111.803 ± 13.409	−14.481 ± 32.140	−165.401 ± 13.960	−168.641 ± 16.706

## Data Availability

All data included in the main text.

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
