# Peer review of "Bio-Guided Isolation of SARS-CoV-2 Main Protease Inhibitors from Medicinal Plants: In Vitro Assay and Molecular Dynamics"

_plants, 2022, doi:10.3390/plants11151914_

Round 1
Reviewer 1 Report
1. The reviewer advises to read the review and refer to it in the introduction and in the text of the article. Abidemi J. Akindele, Abimbola Sowemimo, Foluso O. Agunbiade, Margaret O. Sofidiya, Olufunsho Awodele, Omobolanle Ade-Ademilua, Ifeoma Orabueze, Ismail O. Ishola, Christianah I. Ayolabi, Olumuyiwa B. Salu, Moshood O. Akinleye, Ibrahim A. Oreagba [ACEDHARS UNILAG COVID-19 Response Team]. Bioprospecting for Anti-COVID-19 Interventions From African Medicinal Plants: A Review. Natural Product Communications Volume 17(5): 1–42. 2022.
2. The authors should pay more attention to the botanical component of the information they provide. It is necessary to clarity the data of the Table 1. It is necessary to more clearly distinguish between the belonging of plants to botanical family. The Asteraceae family includes plants 6-10 and 12-14. Plant 11 belongs to Amarantaceae family. More accurate plant name 1 is Barleria trispinosa (Forssk.) Vahl. In Table 1, the intervals in the names of plants should be observed. Number 30 is missing in Table 1, and number 29 appears twice: for OM-1239 and for CG-0440.
1. The Asteraceae family includes plants 6-10 and 12-14. Plant 11 belongs to Amarantaceae family. More accurate plant name 1 is Barleria trispinosa (Forssk.) Vahl. In Table 1, the intervals in the names of plants should be observed. Number 30 is missing in Table 1, and number 29 appears twice: for OM-1239 and for CG-0440.
3. It is necessary to pay more attention to the names of chemical compounds. So, gardenin is named in three different ways (Table 2 and Supplementary Materials).
4. It is necessary to indicate the purity of isolated compounds. The spectra do not always adequately reflect this parameter affecting activity.

Author Response
The reviewer advises to read the review and refer to it in the introduction and in the text of the article. Abidemi J. Akindele, Abimbola Sowemimo, Foluso O. Agunbiade, Margaret O. Sofidiya, Olufunsho Awodele, Omobolanle Ade-Ademilua, Ifeoma Orabueze, Ismail O. Ishola, Christianah I. Ayolabi, Olumuyiwa B. Salu, Moshood O. Akinleye, Ibrahim A. Oreagba [ACEDHARS UNILAG COVID-19 Response Team]. Bioprospecting for Anti-COVID-19 Interventions From African Medicinal Plants: A Review. Natural Product Communications Volume 17(5): 1–42. 2022.
Re: The authors raise their appreciation for the suggested reference that help us in improving introduction and discussion parts.
- The authors should pay more attention to the botanical component of the information they provide. It is necessary to clarity the data of the Table 1. It is necessary to more clearly distinguish between the belonging of plants to botanical family.The Asteraceae family includes plants 6-10 and 12-14. Plant 11 belongs to Amarantaceae More accurate plant name 1 is Barleria trispinosa (Forssk.) Vahl. In Table 1, the intervals in the names of plants should be observed. Number 30 is missing in Table 1, and number 29 appears twice: for OM-1239 and for CG-0440.
Re: The authors revised all plant names and families based on the reviewer’s comments.
- It is necessary to pay more attention to the names of chemical compounds. So, gardenin is named in three different ways (Table 2 and Supplementary Materials).
Re: Chemical name has been revised
- It is necessary to indicate the purity of isolated compounds.The spectra do not always adequately reflect this parameter affecting activity.
Re: Purity charts for isolated compounds are inserted in the supplementary section
Reviewer 2 Report
The manuscript from Abdallah et al. describes the bio-guieded purification of SARS-COV-2 Main Protease Inhibitors from Medicinal Plants. Firstly, the authors evaluated 33 plants extracts for their ability to inhibit SARS-35 CoV-2 MPRO. From the positive hits, major compounds were purified and tested for SARS-35 CoV-2 MPRO. After, the best compounds were evaluated against SARS-CoV-2 40 (hCoV-19/Egypt/NRC-03/2020). The authors performed in silico analysis of the best compounds for explain the obtained results. Overall the manuscript is well presented and organized. It deserves be accepted for publication after minor adjustments. The suggestions are provided in the attached pdf file.
Author Response
We are very pleased with the detailed reviews, and encouraged by the thoughtful comments from the reviewer. Accordingly, point by point response to the reviewer’s comments is presented below.
The manuscript from Abdallah et al. describes the bio-guieded purification of SARS-COV-2 Main Protease Inhibitors from Medicinal Plants. Firstly, the authors evaluated 33 plants extracts for their ability to inhibit SARS-35 CoV-2 MPRO. From the positive hits, major compounds were purified and tested for SARS-35 CoV-2 MPRO. After, the best compounds were evaluated against SARS-CoV-2 40 (hCoV-19/Egypt/NRC-03/2020). The authors performed in silico analysis of the best compounds for explain the obtained results. Overall the manuscript is well presented and organized. It deserves be accepted for publication after minor adjustments. The suggestions are provided in the attached pdf file.
- Line 63-67: Please verify these sentences since the proteases are encoded by the viral genome…
Re: The sentence has been changed to: “The genomes of the SARS- and MERS-CoVs include two open reading frames, ORF1a and ORF1b, which are respectively translated by host ribosomes into two viral polyproteins, pp1a and pp1ab. Two cysteine proteases, a 3C-like protease (3CLpro) and a papain-like protease (PLpro), are encoded by ORF1a. Whereas PLpro cleaves the polyprotein's initial three cleavage sites, 3CLpro cleaves the other 11 sites, releasing a total of 16 non-structural proteins (nsp) for both the SARS-CoV and the MERS-CoV”
- Line 69-71: I think 6LU7 is not an alternative name for this protein, it is the PDB code. Please consider to rewrite this sentence.
Re: The sentence has been changed to: The SARS-CoV-2 3CL (PDB ID: 6LU7), has a 96% resemblance to the SARS-CoV 3CL.
- Please make clear how these compounds were selected.
Re: the authors isolated the known famous major compounds in the extract
- Please consider to include the graph for GC37
Re: the figure has been inserted.
Reviewer 3 Report
Dear Authors,
I appreciate your effort in doing this study. The result of this study can potentially be good but the representation of the study does not reflect that at all. I believe manuscript does not fulfill the standards established for the journal to be considered for publication in its current form.
Moreover, what is the novelty in the manuscript? What new has been added in this manuscript in comparison to plethora of literature available related/similar to the subject? Authors have failed to evident the novelty in the manuscript. Hence, the contribution is weak in this manuscript. This work is saturated and most of the selected compounds have been already explored for their efficacy and 100s of publications are available on scientific engines.
Authors chose 33 plant species. However, only 4 compounds from 33 species (which I believe should have more than 400 bioactive compounds in 33 plants) were selected. What is the basis of selecting only these 4??
In figure 1, why only 7 plants results are portrayed? why not others? Are there inhibition percentage is 0?
What is the minimum percentage of inhibition from figure 1 was considered to be eligible for further tests? Is there any inclusion and exclusion criteria?
I believe authors did the methodology and result section wrong way. The usual way suppose to be; first perform the computational analysis with all major compounds of those 33 plant species, check the stability by molecular dynamics. Once the results from docking and dynamics are satisfactory, then isolate the selected and respective compounds for further in vitro assay. I believe this is the right approach rather than the performing the in vitro assay and confirm with computational study.
Discussion section is unnecessarily long with computational explanations. The major and most important part of discussion is missing. It is suggested to compare the results of the present research with some similar studies which is done before in discussion section. Hence, the contribution is weak in this manuscript. In this manuscript, result and discussion section is basically just results without any discussion. You have made the discussion like another introduction. I would suggest the authors to enhance your theoretical discussion to portray your results in a better way. Authors must compare relevant research work in their discussion.
Author Response
We are very pleased with the detailed reviews and encouraged by the thoughtful comments from the reviewer. Accordingly, point by point response to the reviewer’s comments is presented below.
Dear Authors,
I appreciate your effort in doing this study. The result of this study can potentially be good but the representation of the study does not reflect that at all. I believe manuscript does not fulfill the standards established for the journal to be considered for publication in its current form.
- Moreover, what is the novelty in the manuscript? What new has been added in this manuscript in comparison to plethora of literature available related/similar to the subject? Authors have failed to evident the novelty in the manuscript. Hence, the contribution is weak in this manuscript. This work is saturated and most of the selected compounds have been already explored for their efficacy and 100s of publications are available on scientific engines.
Re: The authors appreciate the reviewer for this valuable comment. There is a lot of publications that used Computational studies to prove the activity of only thymoquinone and gingerol on different target for SARS-CoV-2 like angiotensin-converting enzyme (ACE2) or main protease (MPRO). Other publications discussed activity of the two compounds on the virus itself without assessing the mechanism of action. Meanwhile, no available data regarding other tested compounds 6-Paradol and Gardenin A.
These facts have been clarified in the manuscript as follows:
For thymoquinone: “Previous in silico studies reported the ability of THY to inhibit SARS CoV2 protease [45] as well as ACE2 [46]. Moreover, it has the ability to block the binding of the viral S-protein to the cellular receptor ACE2 of a designed corona virus pseudoparticles, thus blocking viral entry into the host cell. Our results proved for the first time the ability of THY to inhibit SARS COV-2 Viral Main Protease in vitro with IC50 10.26 µM. THY showed a relatively high cytotoxicity and strong anti-SARS-CoV-2 activity with low selectivity index (CC50/IC50=8.2/6.14=1.33).”
For gingerol: Previous reports proved anti- inflammatory,and immunomodulatory properties of the seed [17]. Previous in silico studies and molecular dynamics showed that GNG has binding affinities of –5.60, –5.40 and –5.37 (kcal/mol) against Cathepsin K, COVID-19 main protease, and SARS-CoV 3 C like protease, respectively [49]. Meanwhile, it showed low potency against SARS-CoV-2 infection with IC50 >100 μM (CC50 >100 μM) [50]. In this study GNG was tested for the first time on SARS COV-2 Viral Main Protease invitro with IC50 9.327 µM. Although previous results showed low potency of GNG on the virus [50], but our data showed moderate activity against SARS-CoV-2 virus at non-cytotoxic concentrations in vitro with significant selectivity index (CC50/IC50 = 101.3/43.45 = 2.3).
Meanwhile, PAD was not tested before neither in silico nor in vitro against the six targets for SARS-CoV-2. Although the obtained results showed the highest potency (IC50 0.1682 µM) against SARS COV-2 Viral Main Protease, it showed cytotoxic effect on the tested VERO cells with selectivity index (CC50/IC50 = 155.1/281 = 0.6) which indicated unsuitability of this method to detect its activity.
Gardenin A (GDA) was not tested before for its activity on SARS-CoV-2 and our results are the first report on this compound
- Authors chose 33 plant species. However, only 4 compounds from 33 species (which I believe should have more than 400 bioactive compounds in 33 plants) were selected. What is the basis of selecting only these 4??
Re: In a previous study published by Kanjanasirirat, et al Sci. Rep. 2020, 10, 1-12, the authors studied the effect of 122 Thai natural products for their antiviral activity using fluorescence‐based SARS‐CoV‐2 nucleoprotein detection in Vero E6 cells. From all these plants only Boesenbergia rotunda extract and its phytochemical compound, panduratin A, exhibited the potent anti‐SARS‐CoV‐2 activity.
Therefore, in antiviral assay a researcher may perform screening on one hundred plant extracts and no positive results could be obtained.
The selection of the three plants; Psiadia punctulata, Aframomum melegueta, and Nigella sativa was based on their SARS-CoV-2 MPRO inhibitory activity (more than 30%).
- In figure 1, why only 7 plants results are portrayed? why not others? Are there inhibition percentage is 0?
Re.: only seven plant extracts showed SARS-CoV-2 MPRO inhibitory activity, the other plant extracts did not display any activity (percentage inhibition in Table 1 is marked as “–“).
- What is the minimum percentage of inhibition from figure 1 was considered to be eligible for further tests? Is there any inclusion and exclusion criteria?
Re: The authors appreciate the reviewer for this valuable comment and the following statement has been inserted: “The potent extracts (with inhibition percentage more than 30%) of P. punctulata, A. melegueta and N. sativa were subjected to different chromatographic procedures to isolate their major active constituents.”
- I believe authors did the methodology and result section wrong way. The usual way suppose to be; first perform the computational analysis with all major compounds of those 33 plant species, check the stability by molecular dynamics. Once the results from docking and dynamics are satisfactory, then isolate the selected and respective compounds for further in vitro assay. I believe this is the right approach rather than the performing the in vitro assay and confirm with computational study.
Re: the authors thank the reviewer for his comment, and accordingly we hope to clarify that screening of activity may be performed by two different approaches which could be individually applied based on the study design, aim, and objectives. The first is based on a systematic computer-aided virtual screening approach using commercially available phytoconstituents to identify potential natural product inhibitors of SARS-CoV-2 biological targets. The active hits from the screening will be selected for in vitro evaluation on the viral target (SARS-CoV-2). (as published before by the authors of this manuscript in Pharmaceuticals 14.3, 2021: 213).
The second approach depend on direct testing of plant extract or compound for their inhibitory activity on the virus itself as previously published by Kanjanasirirat, et al Sci. Rep. 2020, 10, 1-12. Then, molecular modelling approach would be applied to investigate the molecular aspects for top-active compound binding towards the examined target. This approach would provide a focused comparative insights for the active metabolites as well as useful directions for future lead optimization and development as possible clinical candidates which suit our study design. Moreover, adopting this approach is perfectly suiting our study since most of the tested plants have not been characterized before. Therefore, preliminary computational analysis is not feasible in our case since performing the computational analysis with all major compounds of those 33 plant species (most of their active constituents are not known) and check the stability by molecular dynamics is considered stochastic with high computational expenses without adding potential advantage over our adopted approach.
Our computational/in vitro testing approach has been reported beneficial for several published reports, not just for SARS-CoV2, but for a variety of biological targets:
- Elokely et al (J Chem Inf Model. 2021;61(9):4745-4757), performed initial in vitro screen of compound collection aiming to identify new lead scaffolds for Mpro, and the obtained hits were investigated via molecular modeling studies to understand the binding characteristics of the identified compounds.
- Cutler et al. (Phytomedicine. 2018;40:27-36), investigated the in vitro inhibition activity of extracts and purified constituents of H. afrum and C. villosus on recombinant human MAO-A and B, and then apply computational protein-ligand docking and molecular dynamics simulations were carried out to explain the MAO binding at the molecular level.
- Hegazy et al. (Biomedicines. 2022;10(5):1169), evaluated the anti-virulence and anti-quorum sensing (anti-QS) activities of dipeptidase inhibitor-4 (DPI-4) antidiabetic gliptins against Gram-negative Pseudomonas aeruginosa and Gram-positive Staphylococcus aureus through initial in vitro assessment of their antibiofilm activities. Identified hits were adopted into a detailed molecular docking study to evaluate the gliptins’ binding affinities to microorganism QS receptors, which helped explain the anti-QS activities of these gliptin hits.
- Discussion section is unnecessarily long with computational explanations. The major and most important part of discussion is missing. It is suggested to compare the results of the present research with some similar studies which is done before in discussion section. Hence, the contribution is weak in this manuscript. In this manuscript, result and discussion section is basically just results without any discussion. You have made the discussion like another introduction. I would suggest the authors to enhance your theoretical discussion to portray your results in a better way. Authors must compare relevant research work in their discussion.
Re: the authors thank the reviewer for his comment, and accordingly we hope to clarify that computational explanation was comprehensive enough to provide clarification for the comparative ligand-Mpro binding. These comprehensive details would useful to understand the binding characteristics of the identified compounds at molecular level for future guidance towards lead optimization and development.
As per-reviewer recommendations, the discussion part has been modified to portray our results in a better way, where computational findings were thoroughly compared with results from current literature (please refer to yellow highlights).
Round 2
Reviewer 3 Report
Manuscript is significantly improved by the authors. However, authors can improve the introduction as well as discussion by comparing similar results as well as importance of medicinal plants for the therapeutic management of COVID-19 from below manuscripts, which can be cited for better impact.
1. Natural products can be used in therapeutic management of COVID-19: Probable mechanistic insights.
2. Identifying the Most Potent Dual-Targeting Compound(s) against 3CLprotease and NSP15exonuclease of SARS-CoV-2 from Nigella sativa: Virtual Screening via Physicochemical Properties, Docking and Dynamic Simulation Analysis
3. Plants-Derived Biomolecules as Potent Antiviral Phytomedicines: New Insights on Ethnobotanical Evidences against Coronaviruses
Author Response
References provided by the respected reviewer have been used to revise and modify the introduction and discussion sections of the manuscript.